# Antimicrobial Activity and Prevention of Bacterial Biofilm Formation of Silver and Zinc Oxide Nanoparticle-Containing Polyester Surfaces at Various Concentrations for Use

**DOI:** 10.3390/foods9040442

**Published:** 2020-04-06

**Authors:** Fabio Fontecha-Umaña, Abel Guillermo Ríos-Castillo, Carolina Ripolles-Avila, José Juan Rodríguez-Jerez

**Affiliations:** Departament de Ciència Animal i dels Aliments, Facultat de Veterinària, Universitat Autònoma de Barcelona, Travessera dels Turons s/n. Bellaterra, 08193 Barcelona, Spain; farwell75@hotmail.com (F.F.-U.); abelguillermo.rios@uab.cat (A.G.R.-C.); carolina.ripolles@uab.cat (C.R.-A.)

**Keywords:** nanoparticles, zinc oxide, silver, antimicrobial, biofilms, polyester, *Escherichia coli*, *Staphylococcus aureus*, *Listeria monocytogenes*

## Abstract

Food contact surfaces are primary sources of bacterial contamination in food industry processes. With the objective of preventing bacterial adhesion and biofilm formation on surfaces, this study evaluated the antimicrobial activity of silver (Ag-NPs) and zinc oxide (ZnO-NPs) nanoparticle-containing polyester surfaces (concentration range from 400 ppm to 850 ppm) using two kinds of bacteria, Gram-positive (*Staphylococcus aureus*) and Gram-negative (*Escherichia coli*), and the prevention of bacterial biofilm formation using the pathogen *Listeria monocytogenes*. The results of antimicrobial efficacy (reductions ≥ 2 log CFU/cm^2^) showed that at a concentration of 850 ppm, ZnO-NPs were effective against only *E. coli* (2.07 log CFU/cm^2^). However, a concentration of 400 ppm of Ag-NPs was effective against *E. coli* (4.90 log CFU/cm^2^) and *S. aureus* (3.84 log CFU/cm^2^). Furthermore, a combined concentration of 850 ppm Ag-NPs and 400 ppm ZnO-NPs showed high antimicrobial efficacy against *E. coli* (5.80 log CFU/cm^2^) and *S. aureus* (4.11 log CFU/cm^2^). The results also showed a high correlation between concentration levels and the bacterial activity of Ag–ZnO-NPs (R^2^ = 0.97 for *S. aureus*, and R^2^ = 0.99 for *E. coli*). They also showed that unlike individual action, the joint action of Ag-NPs and ZnO-NPs has high antimicrobial efficacy for both types of microorganisms. Moreover, Ag-NPs prevent the biofilm formation of *L. monocytogenes* in humid conditions of growth at concentrations of 500 ppm. Additional studies under different conditions are needed to test the durability of nanoparticle containing polyester surfaces with antimicrobial properties to optimize their use.

## 1. Introduction

Bacterial transfer from contaminated food contact surfaces to food products is one of the main emerging public health problems worldwide given that it is a cause of bacterial foodborne outbreaks [1,2]. Bacterial cells in contact with surfaces have the ability to adhere, colonize, and form biofilms that increase the risk of bacterial cross-contamination [3,4,5,6]. To this effect, a variety of cleaning and disinfectant products are used to prevent bacterial growth on surfaces [7]. However, despite the current efforts made by the food industry, bacterial contamination is not easy to remove. In response to the growing concern about bacterial contamination on food contact or other high-risk surfaces, a significant number of studies in the field of chemical or enzymatic engineering have been undertaken. These studies include the development of antimicrobial surfaces capable of preventing and/or inhibiting bacterial adhesion [8,9,10,11]. 

Metal agents such as silver, copper, zinc, titanium, and cobalt have antimicrobial properties. When these agents remain on surfaces, their biocidal properties are gradually released onto the surface through a process of ion exchange, providing continuous, lasting antimicrobial activity [12,13,14]. The biocidal properties of metal agents depend on the characteristics of their nanoparticles, such as size, distribution, morphology, and the release of very small quantities of their cations on the substrates on which they are acting [15,16]. The antimicrobial activity of these agents is also affected by extrinsic factors, such as the pH of the medium, time of exposure, concentration, UV radiation, and the presence of other substances or microorganisms [17,18,19].

Two of the most extended nanocomposites for antibacterial purposes are silver and zinc oxides embedded in polymeric matrices [20,21,22]. The antimicrobial activity of silver (Ag) and zinc (Zn) repels or inhibits the initial attachment of bacteria by either exhibiting an anti-biofouling effect or by inactivating cells in contact with surfaces, causing their death [23,24]. Additionally, the antibacterial properties of these agents reduce the risk of cross-contamination, and therefore the possibility of foodborne disease transmission [13,25]. Silver is the most used element to develop nanomaterials because of its potent antimicrobial activity [26]. The mechanism used by silver nanoparticles to inhibit microorganisms is partially known. Ag-NPs first accumulate on the surface of the bacterial membrane, after which they penetrate the cell, changing the permeability and causing substantial damage [27,28,29]. Zinc oxide (ZnO) has been shown to be one of the most promising metallic nanomaterials. ZnO nanoparticles have received increasing attention as antibacterial agents in recent years because of their stability and safety for humans [16,30,31]. It has been suggested that the mechanism of the antibacterial activity of ZnO-NPs is based on its ability to induce oxidative stress [32]. The Zn^+^ ions released interact with the thiol group of the bacterial respiratory enzymes, increasing the production of reactive oxygen species (ROS) and causing oxidative stress in the bacterial cell. This oxidative stress damages the bacterial membranes, DNA, and mitochondria, resulting in the death of the bacteria [33,34,35]. 

*Staphylococcus aureus* and *Escherichia coli*, representative microorganisms of Gram-positive and Gram-negative bacteria, are capable of adhering, colonizing, and forming biofilms on surfaces. These microorganisms are bacterial standard indicators of food contamination in the food industry and their control indicates the adequate hygiene of manufacturing materials and surfaces in food processing [36,37,38]. *L. monocytogenes* is a foodborne pathogen that can also form biofilms [39]. This bacterium can live on various types of surfaces and in adverse conditions, such as refrigeration temperatures, dryness, low pH, and high salt concentrations [40]. To prevent the bacterial growth and formation of biofilms, different materials with antibacterial properties have been developed. Polymers with antibacterial properties are some of the most used materials inhibiting the adherence of microorganisms and are found in many products, such as plastics, synthetic fibers, and textiles, among others [41,42]. Polymers are not toxic to the environment, and they also have other essential advantages, including enhanced antimicrobial action, ease of processing, and a low cost. Nowadays, they are the fastest growing materials in terms of use because of the numerous advantages they have over other materials. These advantages include physical-chemical parameters, such as hydrophobicity, longevity, biocompatibility, and cationic loading, in addition to specific receptor-mediated interactions which allow the activity of polymers to be adjusted to key structural parameters [22]. Thus, the objective of this study was to determine the bactericidal efficacy of nanoparticles of silver (Ag-NPs) and zinc oxide (ZnO-NPs) both separately and together (Ag–ZnO-NPs), in addition to the control of bacteria growth and biofilm formation using Ag-NPs. They were examined at different concentration levels, incorporated into a polymer matrix (polyester), and then used as a food contact surface. The study was performed using both Gram-positive (*Staphylococcus aureus*) and Gram-negative (*Escherichia coli*) bacterial models. Furthermore, the human pathogen, *Listeria monocytogenes*, was used to test the prevention of bacterial biofilm formation on surfaces.

## 2. Materials and Methods

### 2.1. Test Materials

The antimicrobial surfaces for the tests were hard polyester measuring 50 mm × 50 mm and 10.0 mm thick. The surfaces were provided by a manufacturer dedicated to producing polyester-based materials. These surfaces contained three kinds of biocidal agents: Ag-NPs (spherical-shaped particles ~20 nm in diameter), ZnO-NPs (rod-shaped particles ~20 nm in width and ~100 nm in length and), and Ag–ZnO-NPs. The NPs were embedded homogeneously into the polyester surfaces during manufacturing at different concentrations of parts per million (ppm) (Table 1). The ISO 22196:2011 standard [43] for measuring antibacterial activity on plastics and other non-porous surfaces was used to determine the antibacterial activity. Prior to testing, the polymeric surfaces were sterilized in an autoclave for two cycles at 121 °C ± 2 °C for 15 min and then dried in a forced-air oven at 60 °C for 2 h. Lastly, the surfaces were placed separately in sterile Petri dishes (90 mm) and used in the tests.

### 2.2. Bacterial Cultures and Solutions

Strains of *S. aureus* ATCC 6538 (American Type Culture Collection; Manassas, VA, USA) as Gram-positive bacteria and *E. coli* CECT 515 (Spanish Type Culture Collection, University of Valencia, Paterna, Spain) as Gram-negative bacteria were used for the antimicrobial activity tests, and the strain *L. monocytogenes* CCUG 15526 (Culture Collection University of Gothenburg, Gothenburg, Sweden) was used for the study of prevention of bacterial biofilm formation on surfaces. The strains, obtained as freeze-dried cultures, were recovered by rehydrating them for one hour at 37 °C in buffered peptone water (BPW) (bioMérieux^®^, Marcy-l’Étoile, France), after which they were cultured on tryptic soy agar (TSA; Biokar Diagnostics, Beauvais, France) twice at 37 °C for 24 h. Following the second culture procedure, isolated colonies were used to prepare stock cultures on TSA slants, which were kept for a maximum of one month at 4 °C. The bacterial suspensions of *S. aureus* and *E. coli* for the antimicrobial activity tests were obtained by selecting single bacterial colonies from the stock cultures and incubating them for 24 h at 37 °C on TSA solid plates. After the incubation period, the colonies were transferred from the TSA plates to 9 mL of BPW to obtain the bacterial suspensions in a stationary phase. The concentrations of these bacterial suspensions were then adjusted using a densitometer (Densimat, bioMérieux^®^) to obtain a range between 2.5 × 10^5^ and 1.0 × 10^6^ colony-forming units (CFU)/mL [43]. In the case of *L. monocytogenes*, the bacterial suspensions to study the biofilm formation were obtained by selecting colonies from the stock culture and transferring them to tryptone saline solution (TSS; 1.0 g of tryptone [BD, Madrid, Spain] and 8.5 g NaCl per 1000 mL of sterile distilled water; pH 7.0 ± 0.2).

### 2.3. Antimicrobial Activity Tests

Hard polymeric surfaces containing the antimicrobial agents and polymeric surfaces with no antimicrobial agents as control surfaces were inoculated with 0.4 mL of the bacterial test suspension and covered with a piece of sterile polyethylene film (40 mm × 40 mm) to homogenize the suspension and delimit the contact area. The surfaces were then incubated for 24 h at 37 °C in a humidified chamber (≥90% relative humidity) using pieces of paper towel moistened with sterile water to prevent the bacterial inoculums from drying out [44]. To recover the surviving cells from the polyester surfaces after the incubation period, the samples were transferred into sterile plastic containers with 10 mL of neutralizing solution [1.0 g of tryptone, 8.5 g of sodium chloride, 30 g of Tween^®^ 80, pH 7.0 ± 0.2] containing 8.0 g of glass beads. All the test-neutralizing solutions were then stirred with a shaker under constant agitation at 1000 RPM for 1 min to recover the cells attached to the surfaces. To obtain bacterial counts, dilution series were prepared and cultured in TEMPO^®^ TVC test cards (bioMérieux^®^) for 48 h at 30 °C. After this time had elapsed, the test cards were read with the TEMPO^®^ reader system (bioMérieux^®^).

#### 2.3.1. Assessing the Antimicrobial Activity

The antimicrobial activity was calculated using the following formula:R = U*t* − A*t*(1)
where R is the antibacterial effect expressed in CFU, U*t* is the mean number of bacterial counts obtained from the control samples (CFU/cm^2^), and A*t* is the mean number of bacterial counts from the surface samples with antimicrobial properties (CFU/cm^2^). Thereafter, the bacterial counts were converted into decimal logarithms. Further, the antimicrobial efficacy was determined quantitatively by a reduction equal to or higher than 2 log CFU/cm^2^, which could be recovered from the surfaces.

#### 2.3.2. Validation of Antimicrobial Activity Tests

The TEMPO system was used according to the manufacturer’s instructions. This system is a validated method accepted for the enumeration of mesophilic aerobic microbiota [45]. In parallel, and with the aim of validating the TEMPO^®^ TVC, *S. aureus* ATCC 6538 was cultured by means of the conventional plate count method using TSA. These tests were performed after the cell recovery and by using the control surfaces. 

### 2.4. Detection of L. monocytogenes Biofilm Cells

To reproduce the biofilm formation, the nanoparticle-containing polyester surfaces (Ag-NPs in concentrations of 0, 500, 600, and 800 ppm) and stainless steel disc surfaces (type 304 grade 2B finish, 2 cm in diameter and 1 mm thick) were placed in Petri dishes and inoculated with 30 μL of TSS suspension, containing approximately 3 log viable cells/mL of *L. monocytogenes*. Furthermore, to verify the initial number of cells inoculated on the surfaces, five aliquots of the inoculum were seeded on the TSA plates. The plates were immediately placed into a humidified chamber maintained at a saturated relative humidity of ≥90% using pieces of paper towel moistened with sterile water for 72 h at 22 °C–24 °C to allow biofilm formation [46,47]. The vital staining Live/Dead^®^ BacLight™ bacterial viability kit (Molecular Probes Inc., OR, USA) was used to evaluate the biofilm by direct epifluorescence microscopy (DEM). This kit uses two fluorochromic nucleic acids, SYTO 9 and propidium iodide. SYTO 9 penetrates cells with both damaged and intact membranes. Conversely, the propidium iodide penetrates only the cells with a damaged membrane and reduces the SYTO 9 dye. Therefore, by simultaneously applying the two dyes in adequate proportions, the viable cells with intact membrane fluoresce were stained green, while the dead or injured cells were stained red. After incubating the surfaces for 72 h, they were washed twice with 3 mL of sterile distilled water to remove any free bacterial cells from the biofilms, and immediately stained with 20 μL of the vital staining Live/Dead^®^ BacLight™. The stained samples were left in the dark for 15 min at room temperature. The samples were then analyzed by DEM (Olympus BX51/BX52, Olympus, Tokyo, Japan), equipped with a 100 W mercury lamp (USH-1030L) and a double pass filter (U-M51004 F/R-V2), and coupled to a DP50-CU digital camera (Olympus, Tokyo, Japan). The stained samples were observed with 10×, 20×, and 40× objectives to examine the structure of the biofilm and to evaluate the vital state of the bacterial cells. The initial concentration of *L. monocytogenes*, used to inoculate the samples and form the biofilm, was also determined by DEM. Thereafter, an aliquot of the suspension was stained and observations were made with the 40× objective to determine the number of viable bacteria, performing the count of viable bacteria observed in 10 random fields to estimate the number of viable cells. The images were then analyzed using the AnalySIS 3.2 software (Soft Imaging System, Münster, Germany).

To determine the number of viable bacteria resulting from the biofilm formation, each kind of surface was placed on a sterile flask containing glass beads and 10 mL of neutralizing solution (30 g Tween 80 by 1000 mL of tryptic Soy broth [TSB; Oxoid, Madrid, Spain]). Each flask was then vortexed for 90 s at 40 Hz to dislodge the attached cells from the surfaces. Subsequently, the corresponding decimal dilutions were cultured on TSA agar plates with 6 g/L of yeast extract (TSAYE) for 24 h at 37 °C. Data were expressed as log CFU/cm^2^.

### 2.5. Statistical Analysis

Each test material surface was analyzed three times in quadruplicate (n = 12). The TEMPO system was validated in triplicate in twelve separate experiments (n = 36). The software R version 3.1.0 (R Core Team, Vienna, Austria) was used for the statistical analysis. The data from the cell counts on the various test surfaces were analyzed using the analysis of variance (ANOVA). The Tukey *post hoc* test was used to determine the significance of interactions, where *p* ≤ 0.05 was considered statistically significant.

## 3. Results and Discussion

### 3.1. Bacterial Enumeration on Polymer Surfaces

Twenty-four polymer surfaces as controls were analyzed by means of the TEMPO TVC test cards and conventional plate count using TSA to detect *S. aureus*. The results obtained with the TEMPO TVC (6.34 log CFU/cm^2^) and the plate count methodology (6.35 log CFU/cm^2^) showed no significant differences between them (*p* < 0.05) (Table 2), such that the two methodologies had the same sensitivity to detect *S. aureus*. The TEMPO system is an automated most probable number (MPN), which is based on the measurements of fluorescence method for enumerating aerobic bacteria and specific pathogens in foods [48,49,50]. Further, our results are in accordance with those observed by Katase and Tsumura [51], Blagoeva et al. [52], and Łobacz et al. [53], who compared the results of the TEMPO system with the corresponding methods to enumerate total aerobic bacteria, coliforms, *Enterobacteriaceae*, yeast and mold, *S. aureus*, and lactic acid bacteria. Moreover, comparisons of different methods for quantifying *L. monocytogenes* biofilms have shown that among the indirect counting methodologies, the one that gives the best results is the TEMPO system [54]. Therefore, the obtained results were consistent enough to determine the antimicrobial activity of the surfaces in this study.

### 3.2. Antimicrobial Activity of Ag-NPs, ZnO-NPs and Ag–ZnO-NPs on Polyester Surfaces

The results of the antimicrobial tests on the zinc and silver oxide nanoparticles and the combination of the two agents against *E. coli* and *S. aureus* are shown in Table 3. ZnO-NPs as a sole antimicrobial agent did not show antimicrobial activity higher than 2 log CFU/cm^2^ when it was evaluated at concentrations of 400, 500, and 650 ppm against *E. coli* (reductions of 0.12, 0.25 and 0.83 log CFU/cm^2^, respectively) and *S. aureus* (reductions of 0.17, 0.47 and 0.83 log CFU/cm^2^, respectively). When ZnO-NPs were evaluated at maximum study concentration (850 ppm), an antimicrobial efficacy of 2.07 log CFU/cm^2^ against *E. coli* was observed. These results were statistically significant (*p* < 0.05) when compared to the rest of the concentrations used in this study. In the case of *S. aureus* at the same concentration, however, antimicrobial activity higher than 2 log CFU/cm^2^ was not observed, but there was a reduction of 1.19 log CFU/cm^2^. Our results are in accordance with those reported by Applerot et al. [55] and Pasquet et al. [19], who found that ZnO in aqueous suspensions had stronger bactericidal activity against the Gram-negative bacterium *E. coli* than against the Gram-positive bacterium *S. aureus*. Furthermore, Kim et al. [56] investigated the bactericidal action of Ag-NPs, revealing that *E. coli* is inhibited at a low concentration (3.3 nM), which is ten times lower than the minimum inhibitory concentration in *S. aureus* (33 nM). However, the type of active antimicrobial on polymeric surfaces may influence bactericidal activity depending on the type of microorganism. To this effect, Kanazawa et al. [57], studying several polymeric phosphonium salts, found that the bactericidal activity was higher in *S. aureus* than in *E. coli*. In addition, polymeric phosphonium salt exhibited a higher bactericidal activity by two orders of magnitude than the polymeric quaternary ammonium salt. 

The results obtained when using Ag-NPs showed that all the study concentrations (400 ppm to 850 ppm) had antimicrobial activity above 2 log CFU/cm^2^ for both bacteria evaluated. The highest antibacterial values obtained were 4.90 log CFU/cm^2^ against *E. coli*, and 3.62 log CFU/cm^2^ and 3.84 log CFU/cm^2^ (concentrations of 650 ppm and 850 ppm, respectively) against *S. aureus*. Lower antibacterial activities, which were also considered effective, were 2.35 log CFU/cm^2^ against *E. coli* and 2.38 log CFU/cm^2^ against *S. aureus*, both at 400 ppm. In our study, the action of silver embedded in polyester surfaces was more effective against the Gram-negative *E. coli* than against the Gram-positive *S. aureus*, which is consistent with the results of previous studies. Additionally, Feng et al. [58] found differences when exposing *E. coli* and *S. aureus* cultures to silver nitrate in ionic form by X-ray diffraction microanalysis. These authors suggested that *S. aureus* has a stronger defense system than *E. coli*. As a Gram-positive bacterium, *S. aureus* has a peptidoglycan layer in the cell wall, and there is no visible presence of the nuclear region in the center of the cell where the DNA molecules are randomly distributed. This outer layer of the cell wall protects the cell by preventing silver ions from penetrating into the cytoplasm. Kim et al. [56] and Li et al. [59] also reported greater biocidal efficacy of silver nanoparticles against *E. coli*. This efficacy was attributed to differences in the structure of the cell walls between Gram-negative and Gram-positive bacteria [60]. Other studies support this theory, noting that *S. aureus* tends to be more resistant to adverse conditions and to many types of disinfectants than Gram-negative bacteria [61,62,63]. Additionally, it has been suggested that the cell membrane structure is more complex in Gram-negative bacteria, making it more susceptible to environmental changes [64]. Our study supports this claim because the *E. coli* strain, as a Gram-negative bacteria, was more sensitive than the *S. aureus* strain bacteria.

The synergistic action of 400 to 850 ppm of Ag-NPs with 400 ppm of ZnO-NPs increased the antimicrobial activity of both antimicrobial agents for both bacteria tested. Thus, values higher than 3 log CFU/cm^2^ were observed for all the combined concentrations. Even with *S. aureus*, efficacy was found to be up to 4 log CFU/cm^2^ for the combined action of 400 ppm of ZnO-NPs with 650 ppm or 850 ppm of Ag-NPs (4.39 and 4.80 log CFU/cm^2^, respectively). The results obtained when testing the NPs combination against *E. coli* showed that the activity was significantly higher (*p* < 0.05) when combining 850 ppm of Ag-NPs and 400 of ZnO-NPs, with a bacterial reduction of 5.11 log CFU/cm^2^. A characteristic of the antimicrobial activity of Ag-NPs is the synergistic effect when combined with other natural or synthetic compounds [65]. Cowan et al. [66] indicated that when silver and zinc ions are mixed together at different concentrations and used as a coating agent on stainless steel, zinc has the ability to stabilize the coating. This action produces a slow release of silver ions, allowing for lower silver concentrations and extending the useful life of the coating. In this study, when exposing *S. aureus* and *E. coli* inoculums to silver and zinc NP combinations for 24 and 48 h, reductions greater than 5 log CFU/cm^2^ were achieved. Ji and Zhang [67] found that by incorporating an organic and inorganic antimicrobial compound into the surfaces, their different antibacterial mechanisms were enhanced because of inorganic compounds being more effective to eliminate bacteria. In our study, the enhancing effect of zinc and silver also coincided with an increase in the antimicrobial efficacy. This effect was more evident with the lower concentration levels of Ag-NPs (400 ppm) with ZnO-NPs (400 ppm), increasing efficacy by more than 2 log CFU/cm^2^. Our results also revealed an effective movement of silver and zinc ions toward the surface of the polyester material. This allowed these agents to act homogenously in the presence of bacterial suspensions over the entire area of the surfaces under study. Furthermore, a high correlation between the concentrations of the biocides and microbial efficacy was observed: ZnO-NPs (R^2^ = 0.95 for *E. coli* and R^2^ = 0.99 for *S. aureus*), Ag-NPs (R^2^ = 0.72 for *E. coli* and R^2^ = 0.91 for *S. aureus*), and the mixture Ag–ZnO-NPs (R^2^ = 0.99 for *E. coli* and R^2^ = 0.97 for *S. aureus*) (Figure 1). Consequently, the increased concentration of the biocides enhanced the antimicrobial efficacy for both strains tested, *E. coli* and *S. aureus*. 

One of the major problems caused by chemical antimicrobial agents is multi-drug resistance (MDR) [56]. Consequently, various studies on the effect of Ag-NPs against MDR bacteria have been carried out, showing that Ag-NPs have higher bactericidal efficacy compared to penicillin. In addition, the antimicrobial effect against *E. coli* showed synergistic effects when combined with antibiotics such as amoxicillin, making it an effective system to attack bacteria, including *Pseudomonas aeruginosa*, *E. coli* ampicillin-resistant, *Streptococcus pyogenes* erythromycin-resistant, *S. aureus* methicillin-resistant, and *S. aureus* vancomycin-resistant [68]. This resulted in several studies to determine the systems of microorganism resistance to some heavy metals, such as mercury, arsenic, cadmium, and copper. However, the fundamental resistance mechanisms and the molecular regulation of genes are still not well understood [69]. Although there is no legislation governing the durability requirements for an antimicrobial material, the different types of commercial antimicrobial surfaces must keep their antimicrobial activity for determined periods. 

### 3.3. Bacterial Count by Culture and Evaluation of the Biofilm Formation by DEM

The initial number of the inoculated *Listeria monocytogenes* cells (3 log/cm^2^) showed a high correlation between microscopy and TSA plate count (R^2^ = 0.877), so this quantity of cells inoculated enabled the development of biofilms. The biofilm formation of *L. monocytogenes* on the surfaces after 72 h of incubation reached levels of 5.84 log CFU/cm^2^ and 4.64 log CFU/cm^2^ for stainless steel and polyester without biocide, respectively. Likewise, no differences were found between them (*p* > 0.05), with the formation of biofilms on 100% of the untreated surfaces. On comparing the polyester surfaces containing Ag-NPs, it was observed that the growth of *L. monocytogenes* was inhibited on surfaces treated with Ag-NPs. Thus, the number of bacteria adhered to the surfaces with the lowest concentration (500 ppm) was 0.13 log CFU/cm^2^, with no biofilm formation found (0/18) (Table 4). Additionally, the formation of adherent cells with a concentration of 600 ppm was 2.19 log CFU/cm^2^ with 9/18 of biofilm formation, and 1.01 log CFU/cm^2^ (0/18 biofilm formation) for the concentration of 800 ppm. The prevention of biofilm formation with concentrations of 500 and 800 ppm of Ag-NPs, while a concentration of 600 ppm was only half-effective, could be explained by the relationship between bacterial destruction and antimicrobial concentration not always being linear. Bacterial populations are normally difficult to kill at low concentrations, but if the concentration increases a point is reached where most bacterial growth is killed, while at higher concentrations microorganisms can become difficult to destroy because of the adaptation phenomena. Therefore, it is important to use antimicrobials within optimal concentration ranges, and if these ranges are omitted, the effects may be increased or decreased [70]. Moreover, the ability of *L. monocytogenes* to adhere to common food contact surfaces, such as plastic, rubber, stainless steel and glass, is not well understood [71,72]. However, differences have been observed, both in the extent and speed of adhesion to form biofilms, depending on the types of surfaces [73]. Persistent strains of *L. monocytogenes* may be more resistant to cleaning and disinfection used in the food industry, especially when they are attached to surfaces [74]. One approach to prevent the problem of biofilm formation is the prevention of bacterial adhesion [71]. To modify the surface characteristics of the materials and minimize bacterial adhesion, it is important to know the properties that are decisive in the process. To this effect, hydrophobicity and roughness can be easily altered [75]. Another way to control adhesion is by applying both bactericidal and bacteriostatic antimicrobial agents to surfaces, keeping the microbial population low, and in turn, avoiding the risk of cross contamination [9]. Regarding the evaluation of the samples by DEM, it was observed that the stainless steel and polyester surfaces without biocide had a significant number of viable *L. monocytogenes* cells which were forming biofilms on surfaces (Figure 2A,B). However, little or no presence of *L. monocytogenes* was observed on the surfaces with silver compounds and biofilm formation on most of the surfaces was not detected (Figure 2C). Some studies have described the importance of wet conditions and the availability of nutrients for colonization and the subsequent adhesion of bacteria to surfaces. However, by adding a biocidal compound to the surface structure, favorable conditions for the formation of bacterial biofilms are reduced [76]. Moreover, silver has the ability to attack bacterial cells and destroy them, canceling the adhesion, aggregation, and production of extracellular polymeric substances (EPS) on the surface. It also binds to bacterial DNA and RNA through denaturation and inhibits bacterial replication [77]. Therefore, it is believed that the physical relief of surfaces favors bacterial adhesion, as do surface free energies. Stainless steel surfaces are thus prone to the formation of biofilms because of their great surface free energy, from which their hydrophilic characteristics derive [78]. In our study, it was found that both stainless steel and polyester surfaces without biocide were suitable for the adhesion and formation of biofilms, considering that polyester surfaces are much more porous and suitable for fixing microorganisms. In this sense, Teixeira et al. [75] observed that glass, rubber, and stainless steel surfaces, inoculated with *L. monocytogenes* and *Salmonella* spp., were more prone to bacterial adhesion compared to quartz surfaces, which included a biocidal agent (triclosan) in their composition. 

### 3.4. Factors That Affect the Efficacy of Antimicrobial Agents on Surfaces

Some factors must be considered regarding the antimicrobial efficacy of metal agents, such as the matrix, migration, and relative humidity [46,79]. Incompatibility with the matrix can cause the deterioration and decrease of mechanical and antibacterial properties, respectively. Likewise, the matrix must have free spaces to allow the antimicrobial agent to move or migrate toward the surface of the material in use [14,80,81]. The antimicrobial effect described in this study is caused by the migration of biocidal ions to the surface of polymeric surfaces treated with Ag-NPs and ZnO-NPs. Silver has been considered effective to prevent bacterial growth and the formation of biofilms [60]. The release of silver ions and their migration from these materials to the surface depends on the concentration of the silver component in the polymer matrix [82]. Further, the migration of the antimicrobial agents to the surface depends on the biocide impregnated in the polyester matrix. Polymers are the material of choice because of their specific morphology and their chemical and structural nature, with long polymer chains that enable the incorporation and fine dispersion of particles [83]. It is believed that the microporosity and conformation of the matrix can assist the migration movement of the biocidal agent.

To maximize the properties of an antibacterial agent, it should be well distributed on the surface of the polymer, without forming large aggregates, which drastically reduce antimicrobial efficacy [14,66,80]. Cowan et al. [66] established that when surfaces with antibacterial properties are exposed to humid environments and then allowed to dry, antimicrobial efficacy increases. However, this increase is not attributed to biocidal action but to the desiccation of the microorganisms [46], an increase in the antimicrobial action in dry conditions of use [47], or a lower bacterial growth depending on the type of material [84]. 

In our study, this condition was considered since the surfaces were tested under a relative humidity of more than 90%. Some studies suggest that when mature biofilms are formed on food contact surfaces, strong cleaning and disinfection treatments are required to detach the microorganisms forming the structures [85]. In this regard, preventing biofilm formation in the food industry, including on surfaces that include nanoparticles with a bactericidal effect that prevent initial bacterial adhesion to the surface, could be one of the key alternatives for biofilm control. Nowadays, there are new challenges regarding adding biocides to materials. Coatings usually exhibit functionality for at least ten years, but this is reduced in extreme conditions [86]. Contrary to the expectations, an increase in the concentration of an antimicrobial agent only results in a minimal prolongation of the useful life of the material. There is an environmental impact if antibacterial surfaces are replaced too soon. Given that there is a risk of relatively large quantities of biocides being released into the environment, additional studies are needed on the waste generated by these types of materials. Furthermore, ecological demands and international environmental legislations call for a regulated use of this type of surface, and an approach to limit the number of biocides in materials is essential. Additionally, to prevent bacterial resistance, antibacterial agents in polymers should adapt to a slow release technology to optimize their functionality and durability.

## 4. Conclusions

Polyester surfaces embedded with zinc oxide and silver oxide nanoparticles showed sufficient, controlled levels of NPs released to avoid bacterial adhesion. This effectiveness was dependent on the individual or combined activity of the antimicrobial agents and the concentrations employed in the study. Higher antimicrobial activity results were found with Ag-NPs than with ZnO-NPs when they were evaluated individually at concentrations between 400 and 850 ppm. However, when 400 ppm of ZnO-NPs were combined with each concentration of Ag-NPs, they all demonstrated antimicrobial activity for both *S. aureus* and *E. coli*. When using 850 ppm of Ag-NPs, a high bactericidal activity against *E. coli* and *S. aureus* was reached. Likewise, the use of 500 ppm of Ag-NPs on polyester surfaces prevents the formation of biofilms of the human pathogen *L. monocytogenes*. Our study demonstrates that, unlike separate action, the joint action of Ag-NPs and ZnO-NPs has high antimicrobial efficacy, while the use of Ag-NPs prevents bacterial growth and biofilm formation on polyester surfaces. Additional studies under different conditions are needed to test the durability of surfaces with antimicrobial properties to optimize their use.

## Figures and Tables

**Figure 1 foods-09-00442-f001:**
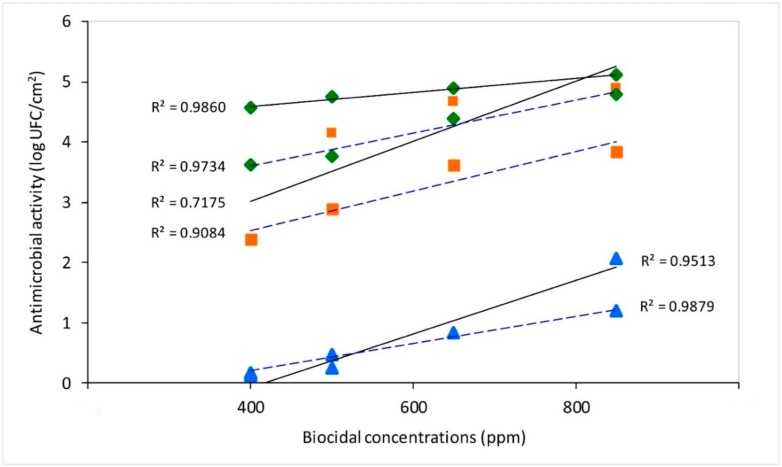
Antimicrobial efficacy against *E. coli* (–––––) and *S. aureus* (– – – –) at different concentrations on biocidal surface types: 
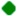
 Ag–ZnO-NPs, 
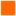
 Ag-NPs, and 
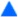
 ZnO-NPs.

**Figure 2 foods-09-00442-f002:**
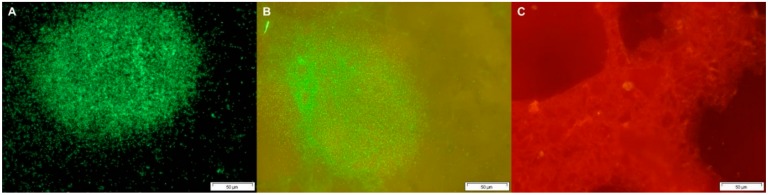
Fluorescence microscopy images of surfaces stained with the Live/Dead^®^ kit. Live cells in biofilms appeared green in color and dead or injured cell appeared red. (**A**) stainless steel (type 304 grade B finish); (**B**) polyester without biocide, forming biofilms of *Listeria monocytogenes* after 72 h of incubation in humid conditions; (**C**) polyester surface treated with 800 ppm Ag-NPs inoculated with *Listeria monocytogenes* after 72 h of incubation in humid conditions, not forming biofilms. All images at 40× magnification.

**Table 1 foods-09-00442-t001:** Concentration (ppm) on nanoparticle-containing polyester surfaces for the study of antimicrobial activity.

Biocidal Agent	Concentration (ppm)
Ag-NPs	400
	500
	650
	850
ZnO-NPs	400
	500
	650
	850
Ag–ZnO-NPs	400 + 400
	500 + 400
	650 + 400
	850 + 400

**Table 2 foods-09-00442-t002:** Comparison of cell counts (log CFU/cm^2^) between TEMPO and conventional plate count.

Number of Readings	Type of Culture	Bacterial Count
24	TEMPO	6.34 ± 0.15 ^a^
24	Plate count	6.35 ± 0.18 ^a^

^a^ The mean values are not significantly different (*p* > 0.05).

**Table 3 foods-09-00442-t003:** Antibacterial efficacy of silver, zinc oxide, and silver–zinc oxide nanoparticles against *Escherichia* coli and *Staphylococcus aureus* on polyester surfaces. Average of the bacterial counts expressed in log CFU/cm^2^ with standard deviation included.

Type of Biocidal Surface	Concentration	Antibacterial Efficacy	
		*E. coli*	*S. aureus*
Ag-NPs	400	2.35 ± 0.46 ^d G^	2.38 ± 0.41 ^c F^
	500	4.14 ± 0.44 ^c F^	2.89 ± 0.33 ^b E^
	650	4.67 ± 0.40 ^b DE^	3.62 ± 0.43 ^a CD^
	850	4.90 ± 0.31 ^a BC^	3.84 ± 0.37 ^a C^
ZnO-NPs	400	0.12 ± 0.51 ^c K^	0.17 ± 0.58 ^d J^
	500	0.25 ± 0.71 ^c J^	0.47 ± 0.83 ^c I^
	650	0.83 ± 0.57 ^b I^	0.83 ± 0.65 ^b H^
	850	2.07 ± 1.01 ^a H^	1.19 ± 0.92 ^a G^
Ag-ZnO NPs	400 + 400	4.57 ± 0.36 ^d E^	3.63 ± 0.33 ^c D^
	500 + 400	4.75 ± 0.37 ^c CD^	3.77 ± 0.41 ^c C^
	650 + 400	4.89 ± 0.36 ^b C^	4.39 ± 0.41 ^b B^
	850 + 400	5.11 ± 0.31 ^a A^	4.80 ± 0.47 ^a A^

^a–d^ Mean values in the same column with different lowercase letters corresponding to each type of biocide are significantly different (*p* < 0.05). ^A–K^ Mean values in the same row with different capital letters are significantly different (*p* < 0.05).

**Table 4 foods-09-00442-t004:** Bacterial growth and biofilm formation of *L. monocytogenes* on polyester and stainless steel after 72 h of incubation under high relative humidity conditions of ≥90%.

Surface	Ag-NPs	Bacterial Growth *	Biofilm Formation
	(Concentration in ppm)		
Stainless steel	0	5.84 ± 0.62 ^a^	18/18
Polyester without biocide	0	4.64 ± 0.92 ^a^	18/18
Poliester A	500	0.13 ± 0.09 ^c^	0/18
Poliester B	600	2.19 ± 2.40 ^b^	9/18
Poliester C	800	1.01 ± 0.60 ^c^	0/17

* Bacterial counts expressed in log CFU/cm^2^. ^a–c^ Average values with different lowercase letters are significantly different (*p* < 0.05).

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
