# Peer review of "Antimicrobial Activity and Prevention of Bacterial Biofilm Formation of Silver and Zinc Oxide Nanoparticle-Containing Polyester Surfaces at Various Concentrations for Use"

_foods, 2020, doi:10.3390/foods9040442_

Round 1

Reviewer 1 Report

This paper describes the evaluation of polymer coated polyester surfaces and their antimicrobial power. In particular silver and zinc oxide nanoparticles in different concentration were evaluated against E. coli and S. aureus to account for Gram- and Gram+ bacteria, and against L. monocytogenes to evaluate prevention of biofilm formation. The article is well written, the introduction very exhaustive, and the topic is relevant in the current scenario of the food science. 

Following my specific comments:

L128 the authors state that the surfaces incubated again, but parameters (time and temperature) of first incubation not stated. Could the authors provide them?

Section 2.3.2. I understand that the authors used TVC cards instead of the specific SA and EC cards specific for S. aureus and E. coli, respectively. They share similar growth requirements. However, why did they choose to validate the system only on S. aureus and not on L. monocytogenes? Of the three it is the one with quite diverse growth behaviour. Also, could the authors provide some more details on the validation?

L267-270 it is not clear what is correlated with the different NPs, could the authors clarify please?

Figure1 Could the authors provide a caption for the different series in the graph (i.e. blue, green, orange)?

L288-290 in the 2.4 section there is no reference to stainless steel and plain polyester, also there is no mention of count after 24h but only of evaluation of LIVE/DEAD cells after 72 hours. 

Table4 How do the authors explain that the intermediate AgNPs concentration allowed more bacterial growth and supported the formation of biofilms in 50% of the samples? Could the author discuss?

Figure2 Could the authors provide more details form figure 2A, like those provided for 2B and 2C?

Line316 EPS appears for the first time here, could the authors provide the full name for clarity? 

Author Response

Reviewer #1:

This paper describes the evaluation of polymer coated polyester surfaces and their antimicrobial power. In particular silver and zinc oxide nanoparticles in different concentration were evaluated against E. coli and S. aureus to account for Gram- and Gram+ bacteria, and against L. monocytogenes to evaluate prevention of biofilm formation. The article is well written, the introduction very exhaustive, and the topic is relevant in the current scenario of the food science. 

RESPONSE:

We really appreciate your comments to improve the quality of the present manuscript. Likewise, in order to improve the writing, once the changes were made, a native English-speaking reviewer reviewed the full manuscript.

Following my specific comments:

L128 the authors state that the surfaces incubated again, but parameters (time and temperature) of first incubation not stated. Could the authors provide them?

RESPONSE:

To clarify, the time and temperature has been added to the sentence (Lines 131-133): “The surfaces were then incubated for 24 hours at 37 °C in a humidified chamber (≥ 90% relative humidity) using pieces of paper towel moistened with sterile water to prevent the bacterial inoculums from drying out”.

Section 2.3.2. I understand that the authors used TVC cards instead of the specific SA and EC cards specific for S. aureus and E. coli, respectively. They share similar growth requirements. However, why did they choose to validate the system only on S. aureus and not on L. monocytogenes? Of the three it is the one with quite diverse growth behaviour. Also, could the authors provide some more details on the validation?

RESPONSE:

Although both S. aureus and E. coli were used for the study about antimicrobial efficacy, S. aureus was chosen to validate the Tempo because it was the most resistant to nanoparticles. However, in previous tests, Tempo was found to be equally effective in counting E. coli (data not shown).

In the case of the prevention of biofilm formation using Listeria monocytogenes, the validation was carried out by comparing the counts obtained by microscopy with those obtained by plate counting. It is described in Materials and Methods (Lines 155-160): “To reproduce the biofilm formation, the nanoparticle-containing polyester surfaces (Ag-NPs in concentrations of 0, 500, 600 and 800 ppm) and stainless steel disc surfaces (type 304 grade 2B finish, 2 cm in diameter and 1 mm thick) were placed in Petri dishes and inoculated with 30 μL of TSS suspension, containing approximately 3 log viable cells/mL of L. monocytogenes. Furthermore, to verify the initial number of cells inoculated on the surfaces, five aliquots of the inoculum were seeded on TSA plates”.

Additionally, the results of the correlation (R2) of both techniques was added to Results and Discussion to give information about the validation (Lines 303-305): “The initial number of Listeria monocytogenes cells inoculated (3 log/cm2) measured by showed a high correlation between microscopy and TSA plate count (R2 = 0.877), so this quantity of cells inoculated enabled the development of biofilms”.

L267-270 it is not clear what is correlated with the different NPs, could the authors clarify please?

RESPONSE:

In our study, it was found that the increase in biocides resulted in greater antimicrobial efficacy for S. aureus and E. coli. Thus, a high correlation was observed between the concentration of biocides and antimicrobial activity. To clarify this description, the phrase was re-written (Lines 282-286): “Furthermore, a high correlation between the concentrations of the biocides and microbial efficacy was observed: ZnO-NPs (R2 = 0.95 for E. coli and R2 = 0.99 for S. aureus), Ag-NPs (R2 = 0.72 for E. coli and R2 = 0.91 for S. aureus), and the mixture Ag–ZnO-NPs (R2 = 0.99 for E. coli and R2 = 0.97 for S. aureus) (Figure 1). Consequently, the increased concentration of the biocides enhanced the antimicrobial efficacy for both strains tested, E. coli and S. aureus”.   

Figure1 Could the authors provide a caption for the different series in the graph (i.e. blue, green, orange)?

RESPONSE:

Corresponding symbols were added to the caption of Figure 1: Ag–ZnO-NPs (rhomb in green), Ag-NPs (square in orange), and  ZnO-NPs (triangle in blue).

L288-290 in the 2.4 section there is no reference to stainless steel and plain polyester, also there is no mention of count after 24h but only of evaluation of LIVE/DEAD cells after 72 hours. 

RESPONSE:

This observation was corrected. It is 72 hours as is described in the caption of Figure 2. The corresponding change was made in Lines 305-307: “The biofilm formation of L. monocytogenes on the surfaces after 72 hours of incubation reached levels of 5.84 log CFU/cm2 and 4.64 log CFU/cm2 for stainless steel and polyester without biocide, respectively”.

Additionally, the information of incubation time of 72 h and the relative humidity was added to Table 4: “Bacterial growth and biofilm formation of L. monocytogenes on polyester and stainless steel after 72 hours of incubation under high relative humidity conditions of ≥ 90%”.           

Table4 How do the authors explain that the intermediate AgNPs concentration allowed more bacterial growth and supported the formation of biofilms in 50% of the samples? Could the author discuss?

RESPONSE:

In our study, the prevention of biofilm formation with concentrations with 500 and 800 ppm of Ag-NPs was effective, while a concentration of 600 ppm was only 50% effective. These results may be due to the fact that the relationship between bacterial destruction and antimicrobial concentration is not always linear. Bacterial populations are normally difficult to kill in low concentrations, but if the concentration increases, a point is reached where most bacterial growth is killed and at higher concentrations, microorganisms can become difficult to destroy by adaptation phenomena. Thus, according to the observations of Holah et al. 1995 (Added to the Reference section), it is important to use antimicrobials in optimal concentration ranges, and if these ranges are omitted, they may increase the effects or decrease their activity. This information has been added to the manuscript in Lines 314-321

Figure2 Could the authors provide more details form figure 2A, like those provided for 2B and 2C?

RESPONSE:

The characteristics of the stainless steel used was added to the Figure 2A: “2A) stainless steel (type 304 grade 2B finish)”, and to the text (Lines 156-157): “…stainless steel disc surfaces (type 304 grade 2B finish, 2 cm in diameter and 1 mm thick)..”.

Line316 EPS appears for the first time here, could the authors provide the full name for clarity? 

RESPONSE:

The full name has been added (Lines 340-341).  

Reviewer 2 Report

One of the concern to use silver nanoparticles is the migration of silver ions into the foodstuff.

  1. I was wondering which is the experimental procedure used to functionalize the surface with NPs: physical interactions or chemical interactions with the polymer. In addition, the NPs are embedded into the polymer or only on the surface. The impact of the manuscript would increase including surface characterization such as SEM images (or at least bright light confocal microscope).
  2. I was wondering if the authors have any information about the size and shape of the nanoparticles as well as surface distribution of these nanoparticles.
  3. the difference of the bacteria growth is not well explained in the manuscript. These could be attributed to the degree of aggregation of the nanoparticles, which can affect the silver ions release. This aggregation can increase according with silver NPs concentration.  For that a characterization of the NPs distribution on the surface is important. This also could be study via the quatification of the ions release from NPs. This could be do by ICP-MS.

Minor revision:

  1. Format of Table 1, Table 3 should be improved adding a line between the NP type.
  2. First sentence of the conclusiones "Polyester surfaces embedded with zinc and silver oxide..." this sentence should be changed: ...zinc oxide and silver...
  3. The authors should include the funding information.

Author Response

Reviewer #2:

One of the concern to use silver nanoparticles is the migration of silver ions into the foodstuff.

RESPONSE:

We appreciate your comments to improve the quality of the present manuscript. Likewise, once the changes were made, a native English-speaking reviewer reviewed the full manuscript.

  1. I was wondering which is the experimental procedure used to functionalize the surface with NPs: physical interactions or chemical interactions with the polymer. In addition, the NPs are embedded into the polymer or only on the surface. The impact of the manuscript would increase including surface characterization such as SEM images (or at least bright light confocal microscope).

RESPONSE:

The authors agree that both physical and chemical interactions are involved in the migration and therefore, the antimicrobials activity of NPs on surfaces. In the case of our study, the main objective was focused on the antimicrobial capacity and the prevention of the formation of biofilms by microbiological methods, considering the concentrations of the NPS. In response to the reviewer's comment, we have added the type of distribution of the NPs on the surfaces, as well as their size and shape in Lines 96-99. Additionally, this study discusses the migration of NPs on surfaces in Lines 363-376.

  1. I was wondering if the authors have any information about the size and shape of the nanoparticles as well as surface distribution of these nanoparticles.

RESPONSE:

The information was added to the text in Lines 96-99: “These surfaces contained three kinds of biocidal agents: Ag-NPs (spherical-shaped particles ~20 nm in diameter), ZnO-NPs (rod-shaped particles ~20 nm in width and ~100 nm in length), and Ag–ZnO-NPs. The NPs were embedded homogeneously into the polyester surfaces during manufacturing at different concentrations of parts per million (ppm)”.

  1. the difference of the bacteria growth is not well explained in the manuscript. These could be attributed to the degree of aggregation of the nanoparticles, which can affect the silver ions release. This aggregation can increase according with silver NPs concentration.  For that a characterization of the NPs distribution on the surface is important. This also could be study via the quatification of the ions release from NPs. This could be do by ICP-MS.

RESPONSE:

The aggregation of the NPs was homogeneously distributed on the surfaces to ensure an adequate migration during their useful life to avoid adherence and bacterial growth. In this sense, high correlations were found between the concentrations of use of NPs and the antimicrobial efficacy for S. aureus and E. coli, described in Lines 280-286: “This allowed these agents to act homogenously in the presence of bacterial suspensions over the entire area of the surfaces under study. Furthermore, a high correlation between the concentrations of the biocides and microbial efficacy was observed: ZnO-NPs (R2 = 0.95 for E. coli and R2 = 0.99 for S. aureus), Ag-NPs (R2 = 0.72 for E. coli and R2 = 0.91 for S. aureus), and the mixture Ag–ZnO-NPs (R2 = 0.99 for E. coli and R2 = 0.97 for S. aureus) (Figure 1). Consequently, the increased concentration of the biocides enhanced the antimicrobial efficacy for both strains tested, E. coli and S. aureus”.

However, when the study was carried out in the prevention of biofilms, it was seen that the increase was not always linear (Lines 314-321): “The prevention of biofilm formation with concentrations of 500 and 800 ppm of Ag-NPs, while a concentration of 600 ppm was only half-effective, could be explained by the relationship between bacterial destruction and antimicrobial concentration not always being linear. Bacterial populations are normally difficult to kill at low concentrations, but if the concentration increases a point is reached where most bacterial growth is killed, while at higher concentrations microorganisms can become difficult to destroy due to adaptation phenomena. Therefore, it is important to use antimicrobials within optimal concentration ranges, and if these ranges are omitted, the effects may be increased or decreased activity [70]”. Similarly, more studies are needed to determine how NPs act on antimicrobial activity at physical-chemical levels, which can be performed using techniques such as ICP-MS or X-ray spectrometry, with the aim of improving the protection of surfaces.

Minor revision:

  1. Format of Table 1, Table 3 should be improved adding a line between the NP type.

Response:

The lines to separate NPs types were added in Table 1 and Table 3.

  1. First sentence of the conclusiones "Polyester surfaces embedded with zinc and silver oxide..." this sentence should be changed: ...zinc oxide and silver...

Response:

This change has been introduced to the text in Line 401.

  1. The authors should include the funding information.

Response:

The funding information has been added (Lines 420-421): “Funding: This research was funded by the Spanish Ministerio de Ciencia, Innovación y Universidades, grant number RTI2018-098267-R-C32”. 

Round 2

Reviewer 2 Report

The revised manuscript has included more information about the physico-chemical properties of the antibacterial surface, which improve the quality of the work. I'm still missing morphological characterization of the NP-modified surfaces because the authors dont show any experimental evidence about the mentioned homogeneity of the NP distribution. However, I can understand that the authors dont have access to electron microscopy for example.

The current manuscript version has enough quality to be published on Foods.